# Organic vs. Conventional Farming of Lavender: Effect on Yield, Phytochemicals and Essential Oil Composition

Ana Dobreva [1], Nadezhda Petkova [2], Mima Todorova [3,*], Mariya Gerdzhikova [3], Zornitza Zherkova [4] and Neli Grozeva [4]

[1] Department of Aromatic and Medicinal Plants, Institute for Roses and Aromatic Plants, Agricultural Academy, 6100 Kazanlak, Bulgaria; anadobreva@abv.bg

[2] Department of Organic and Inorganic Chemistry, University of Food Technology, 4002 Plovdiv, Bulgaria; petkovanadejda@abv.bg

[3] Department of Plant Breeding, Faculty of Agriculture, Trakia University, 6000 Stara Zagora, Bulgaria; m_gerdjikova@abv.bg

[4] Department of Biological Sciences, Faculty of Agriculture, Trakia University, 6000 Stara Zagora, Bulgaria; n.grozeva@trakia-uni.bg (N.G.)

* Correspondence: mima.todorova@trakia-uni.bg; Tel.: +359-42699499

**Abstract:** Increasing prices and market demand for organic products are stimulants of organic farming. However, this sector is a challenge for producers and further improvements are still necessary. The present study case was conducted to compare the effects of organic (OF) and conventional (CF) farming on lavender (*Lavandula angustifolia* Mill.) oil yield, plant pigments and essential oil composition. The study was conducted for two years in the period 2019–2020. Six private farms were included in the experiment with conventional and organic agriculture systems. They are located in Kazanlak Valley, Southern Bulgaria. Organic lavender inflorescences were determined to have chlorophyll a and total chlorophyll within a narrow range between 251.3 and 275.7 $\mu g \cdot g^{-1}$ and between 375.5 and 487.0 $\mu g \cdot g^{-1}$ compared to conventional ones—between 245.9 and 377.5 $\mu g \cdot g^{-1}$ and 385.3 and 595.4 $\mu g \cdot g^{-1}$ respectively. However, carotenoids and anthocyanins were in a wide range in organic lavender between 36.9 and 72.2 $\mu g \cdot g^{-1}$ and 410 and 1240 $\mu g$ cyn-3-gly·$g^{-1}$ compared to conventional ones—between 55.5 and 77.3 $\mu g \cdot g^{-1}$ and 200 and 780 $\mu g$ cyn-3-gly.$g^{-1}$, respectively, for both studied years. The key constituents in essential oil were linalyl acetate (28.42–38.23%), linalool (20.01–31.04%) and β-caryophyllene (7.95–14.97%). The composition was compared with the parameters set out in the international standard for lavender oil. The influence of the type of agricultural system on essential oil yield and its composition was not found. According to the obtained results, levels of chlorophyll a, chlorophyll b and total chlorophyll were higher in conventional farming than in organic farming for the second year of the study.

**Keywords:** *Lavandula angustifolia* Mill.; agriculture system; plant pigments; quality; volatiles

## 1. Introduction

Lavender is a perennial semi-shrub plant from the Lamiaceae family, genus *Lavandula* L. The family includes more than 30 botanical species [1,2], of which the following 3 species are of economic importance: *Lavandula angustifolia* Mill. (lavender), *Lavandula intermedia* Emeric ex Loisel. (lavandin) and *Lavandula latifolia* L. (spike) [3,4]. Lavandula angustifolia Mill. has been used for medicinal purposes since ancient times, and nowadays lavender essential oil from Lavandula angustifolia Mill. is one of the most valuable oils in the world. Its successful use in perfumery, medicine and pharmacy is due to the presence of antiseptic, pain-relieving, antispastic, bactericidal, sedative and antioxidant properties that the essential oil possesses. In some countries, lavender is also used as a spice in dishes [5]. *Lavandula angustifolia* Mill. is grown mainly in the temperate regions of Europe, Asia, America and Australia. For the conditions of Bulgaria, it is one of the main essential oil

crops. The history of lavender in our country dates back to the beginning of the last century (1907), when this essential oil crop was imported into the country. In the beginning, it was grown in the area of the so-called Rose Valley, where the first Bulgarian varieties were cultivated in the 1960s. A little later, the varieties "Hemus", "Sevtopolis", "Druzhba" and "Yubileyna" were developed, which today are among the most common in our country. In recent decades, due to the enhanced interest in lavender production, cultivation areas in various regions of the country have increased, and today, Bulgaria has become a world leader with a production of 400 tons per year [2,3].

In recent years, there has been a growing interest in the production and processing of organic agricultural products, which in turn has led to an increase in agricultural crops grown through organic agriculture compared to conventional agriculture [6–8]. Organic agriculture is a production system that does not allow or completely excludes the use of synthetic fertilizers, pesticides or growth regulators, in which crop rotations, plant residues, organic fertilizers, green fertilization as well as biological plant protection to deal with various diseases and pests are relied on to maintain and improve the nutritional regime of the soil. Nowadays, organic farming is seen as a system that is much more than excluding the use of pesticides and mineral fertilizers. It is a different production method that has a holistic nature, involving the application of many preventive measures. The choice of organic farming for an essential oil crop compared to the conventional one is often registered as more expensive and risky for agricultural production, but the results are of particular importance for a better lifestyle, health and longevity [9]. Organic farming can be described as a form of agriculture that uses natural resources in a sustainable way and strategies such as the application of biofertilizers, biological pest control, crop rotations [10], timely forecasting and signaling as well as ecologically sound methods aimed at preserving and increasing soil fertility.

Lavender can be grown on poorly fertile soils and sloping terrains, helping to protect them from erosion and at the same time being a source of high income. The fact that it is also a valuable honey plant makes it extremely suitable for regions where the cultivation of more demanding crops is limited. Lavender has another advantage: it is also a drought-resistant plant, adapted to use moisture from the deep soil layers.

Nowadays, the cultivation of lavender can be carried out through the typical conventional type of farming as well as through organic farming. For essential oil crops used in medicine, the quality of the final product is extremely important. In this regard, there are a number of studies related to the quality of production of medicinal plants, depending on the applied agricultural practices when growing the plants [11,12]. Essential oil composition is determined by the plant genotype, but abiotic factors and growing conditions can also affect biomass production and oil quality. For example, the team of Todorova et al. (2022) [13] found that the type of agrarian production system has no influence on the chemical composition of the rose oil from the R. Damascena oil-bearing plant, but the highest productivity of essential oil was obtained in combined ecological agricultural practices (low-input farming system)—0.039%, followed by typical conventional farming—0.036% and the lowest productivity was obtained from rose plantations grown through organic farming—0.030%. The study by Ghrysargyris et al. (2016) [14] in hydroponic cultivation of *L. angustifolia* Mill. on the effect of mineral fertilization with nitrogen in doses (N: 150–175–200–225–250 mg/L) and phosphorus (P: 30–40–50–60–70 mg/L) on the morphological and biochemical characteristics of lavender concluded that phosphorus—P fertilization doses mainly affect the growth of the plants themselves, and the amount of essential oil remains unaffected by the various rates of N and P fertilization. The studies by Yassemin et al. (2017) [15] with the following levels of nitrogen fertilization (0, 50, 100, 200, 400 and 800 mg/L N; $NH_4NO_3$) of *Lavandula angustifolia* Mill. led to the conclusion that fertilization doses significantly affected the development of root length, stem and root neck thickness, as well as leaf chlorophyll concentration. The different mineral fertilizers used, types of pesticides and their relationship with volatile chemicals continue to be the subject of research [16], as well as the question of whether the type of agriculture, organo-mineral

fertilization, has an effect on the amount and composition of lavender essential oil [12]. Organo-mineral fertilization with an appropriate harvest season has a noticeable effect on the chemical composition of lavender oil, especially on 1,8-cineole and phenone [17]. Renaud et al. [11] compared the agronomic characteristics, quantity and quality of essential oil of 10 cultivars of certified organic lavender (*Lavandula* spp.), finding that the enantiomeric distribution of (R)-(−) and (S)-(+) forms of linalool and linalyl acetate are useful indicators of the purity of lavender oils. Consumer demand for lavender products has increased in favor of organically produced ones, with particular attention paid to the purity of lavender oils [2].

On the other hand, the measurement of plant pigments, such as chlorophyll a and b, carotenoids and the ratio between them, measured in leaf mass, allow for an express assessment of the physiological state of plants and can be used as indicators of abiotic, biotic or anthropogenic stress in plants [18].

The objective of our study was to compare the oil yield, phytochemicals and essential oil composition of lavender in organic and conventional farming systems to find out the impact of this farming system on the productivity and quality of the essential oil.

## 2. Materials and Methods

The field study was performed on six private farms located in the Kazanlak Valley, Bulgaria, during the two-year period of 2019–2020. The valley is located at 400–500 m a.s.l., in the middle of the country between Stara Planina mountain to the north and Sredna Gora mountain to the south. The climate is continental and relatively mild, with warm winters and cool summers. Winters are warmer and summers cooler. The average annual temperature in the area is around 11 °C and annual precipitation is 540 mm. Spring is also cool and rainy. The study area location is presented in Figure 1.

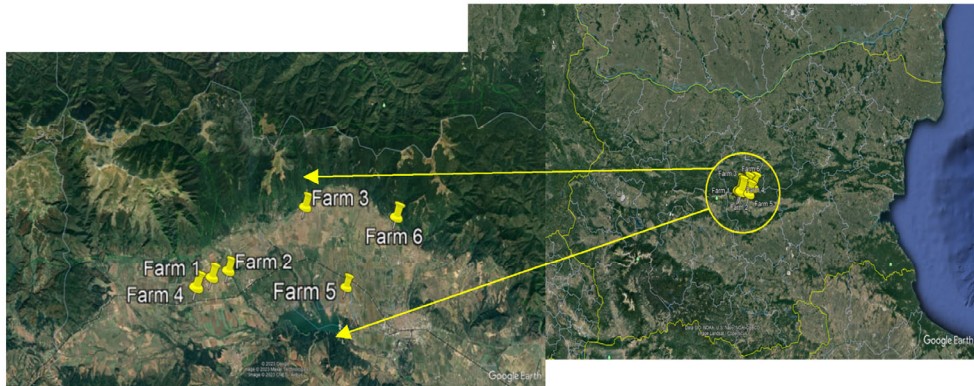

**Figure 1.** Study area location—Google Earth.

The farms are located close to each other, where the main soil type is Fluvisols. Three of the plantations are certified as organic farming (OF) and the rest are characterized as conventional farming (CF). The geographic coordinates of each study area were measured with a Garmin GPS. Soil samples were collected from each zone from the 0–30 cm surface horizon using Eijkelkamp soil sampling equipment. Soils were air-dried, ground and sieved with particles below 2 mm. The samples were analyzed for some soil parameters such as organic matter content by loss on ignition and pH values by the potentiometric method. The private farms in the present study are located close to one another, for example, the distance between Farm 1 (Asen) and Farm 2 (Asen) is less than a kilometer and both are owned by the same owner. Farm 3 (Yasenovo) is located 5 km east of farms 1 and 2. The distance between Farm 3 and Farm 6 (Krun) is 10 km. The longest is the distance between Farm 4 (Gabarevo) and Farm 5 (Koprinka)—14 km. The plantations of *L. angustifolia* Mill. in all six fields were created according to the generally accepted technology for the country with an inter-row distance of 1.4 m and within the row distance of 0.35–0.40 m. The study area is

known for the cultivation of essential oil crops, such as oil-bearing rose and lavender. Three varieties of *L. angustifolia* are grown in the area—Hemus var., Sevtopolis var. and Yubileyna var. The Hemus variety is characterized by very stable yields even under unfavorable growing conditions. The private farms selected by us cultivate mainly the Hemus variety without irrigation. Detailed information on the agricultural practices carried out in the studied private fields is presented in Table 1. The soil tillage included 3–4 hoeings with a cultivator between the rows and in the row on all fields as hand hoeing in the row was applied in Farms 1 and 2. The management practices were carried out every year of the study. In general, the main differences between agriculture technologies in OF and CF are mainly related to the type of fertilization and pest control. In our study case, mineral fertilization with NPK (YaraMila COMPLEX from KVS Agro, Sofia, Bulgaria) with a dose of 150–200 kg/ha was applied before vegetation and one-time foliar feeding with NPK + microelements during the vegetation in CF farms at the end of April. The systemic soil herbicide Devrinol (napropamide) was applied at a dose of 4 L/ha before the beginning of plant development in all CF and insecticide Deca EK 60 mL/da against locust attack in Farm 6 (CF). Regarding organic farms, manure was applied before the growth season in the three farms. Composted cattle manure with the following composition was used: 0.90% total N; 0.04% inorganic N; 0.31% total P; 10.2% total C and 35.2% water content. In Farm 3 (OF) in 2015, lime reclamation was also applied to correct acid soil reaction. The fields have not been attacked by insects and diseases since the start of lavender planting in both OF and CF. In this regard, neither biological plant protection in OF nor conventional plant protection in CF have been carried out except for Farm 6, where 60 mL/da of Deca EK insecticide was used against locust attack in 2018.

The data concerning the climate conditions in the studied years and the harvest were provided by the local meteorological station in Kazanlak. The monthly distribution of the mean temperatures and precipitation, relative to 100-year mean values (Norms), is presented in Figures 2 and 3, respectively.

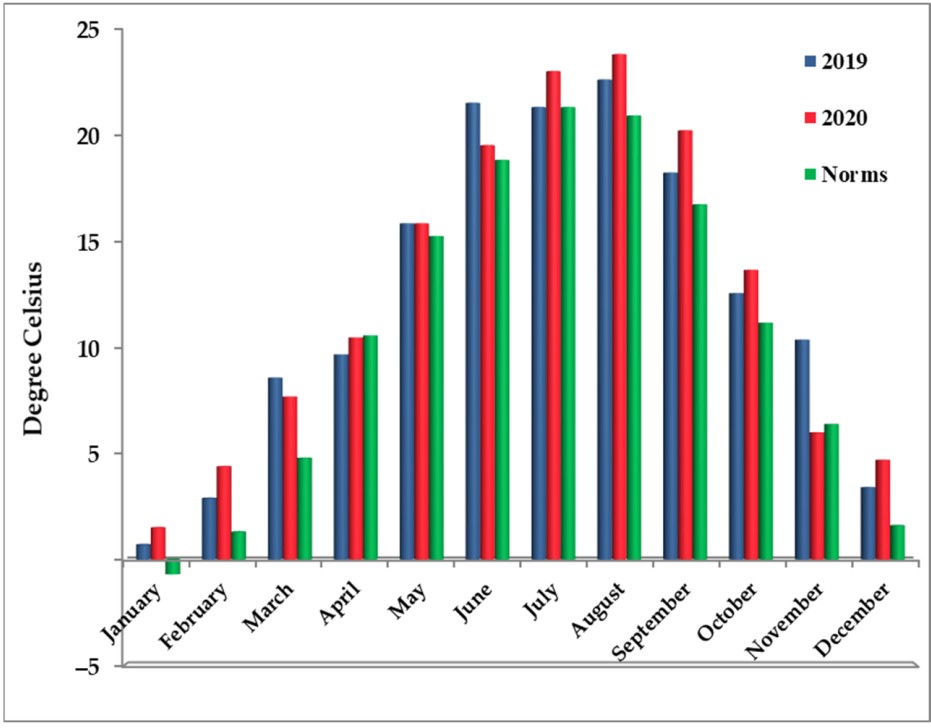

**Figure 2.** The average monthly temperatures (°C) for Kazanlak valley in 2019/2020.

**Table 1.** Farming system, geographical data, general agricultural practices in the studied farms.

| Farms | Location | Localization (Garmin) | Farming Type Since | Altitude, m | Agricultural Practices during Vegetation | Fertilization/Plant Protection/Irrigation |
|---|---|---|---|---|---|---|
| Farm 1 (OF) | Asen | 42°38.548′ N 25°10.590′ E | Organic, 2016 | 483 | Soil tillage: 3–4 hoeing with a cultivator between the rows. Hand hoeing in the row. | 25 t/ha cattle manure was applied No plant protection was applied. No irrigation. |
| Farm 2 (OF) | Asen | 42°38.831′ N 25°11.675′ E | Organic, 2016 | 482 | Soil tillage: 3–4 hoeing with a cultivator between the rows. Hand hoeing in the row. | 25 t/ha cattle manure was applied No plant protection was applied. No irrigation. |
| Farm 3 (OF) | Yasenovo | 42°41.536′ N 25°16.786′ E | Organic, 2011 | 500 | Soil tillage: 3–4 hoeing with a cultivator between the rows and into the rows. | 30 t/ha cattle manure was applied. Lime melioration 3 t/ha No plant protection was applied. No irrigation. |
| Farm 4 (CF) | Gabarevo | 42°38.091′ N 25°09.531′ E | Conventional, 2016 | 423 | Soil tillage: 3–4 hoeing with a cultivator between the rows. | Spring—combined NPK April—foliar fertilizer NPK + micro elements. No plant protection was applied. No irrigation. Devrinol 4 F 400 mL/da (napropamide) was applied. |
| Farm 5 (CF) | Koprinka | 42°38.264′ N 25°19.506′ E | Conventional, 2016 | 395 | Soil tillage: 3–4 hoeing with a ″cultivator with suns″ between the rows. | Spring—combined NPK April—foliar fertilizer NPK + micro elements. No plant protection was applied. No irrigation. Devrinol 4 F 400 mL/da (napropamide) was applied. |
| Farm 6 (CF) | Kran | 42°41.076′ N 25°22.820′ E | Conventional, 2017 | 440 | Soil tillage: 3–4 hoeing with a cultivator between the rows. Hand hoeing in the row. Devrinol 4 F 400 mL/da was applied. | Spring—combined NPK April—foliar fertilizer NPK + trace elements. No irrigation. Devrinol 4 F 400 mL/da, Deka EC 60 mL/da. |

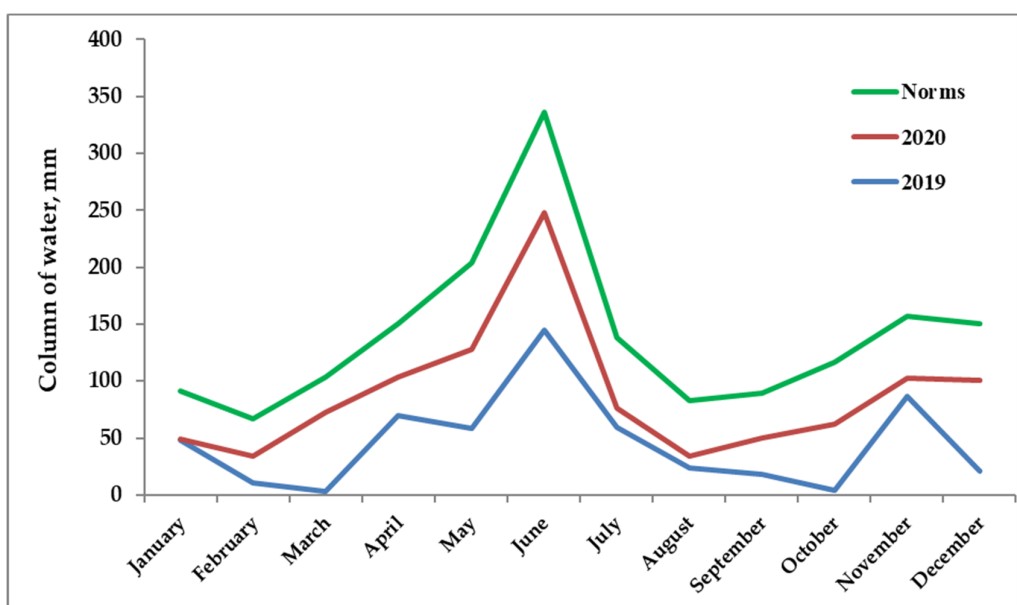

**Figure 3.** The average monthly rainfall (mm) for Kazanlak valley in 2019/2020.

*2.1. Plant Material and Processing*

Lavender inflorescences were used as plant material. They were harvested during the warm hours of the day on 27 June (2019) and 2 July (2020) at the appropriate flowering rate between 50 and 100%, with stems up to 10 cm. The most suitable period for harvesting and sampling is sunny and windless weather [19]. The harvest was conducted on the same day. From each individual field (farm), three samples were taken manually at random, and each of them was collected from about 40 different tuffs, which means 6 fields (3 OF and 3 CF) with 3 replicates, with total samples *n* = 36 for the two years of investigation. Each sample of lavender (1000 g) was split into two parts. The first part was used for distillation and essential oil production. The second part was used for biochemical analysis.

The essential oil was obtained by steam distillation of the material, using a semi-industrial stainless steel set (V = 5 dm$^3$), coupled with a cooler and glass separator. The process parameters were as follows: sample amount, 200 g; distillation rate, 8–10 mL/min; temperature of the distillate, 30 °C and duration of the process, 1.0 h. The essential oil was measured to the graduated part of the separator in milliliters and was calculated as a percentage by volume (*v*/*w*). For better accuracy, a relative density recalculation was made and was presented as a percentage by weight (*w*/*w*). After collection, the oil was treated with anhydrous Na$_2$SO$_4$ and stored in tightly closed vials at 4 °C till analysis.

*2.2. Chemical Analysis*
2.2.1. Plant Pigments

The determination of chlorophyll a (Chla), chlorophyll b (Chlb), total chlorophylls and total carotenoids was performed using the method of Lichtenthaler and Wellburn [20]. The sample was mixed with 80% acetone (1:10 *w*/*v*) and extracted in an ultrasonic bath SIEL UST 5.7-150 at 40 °C for 15 min. The extraction was performed in triplicate. The combined acetone extracts were measured at three wavelengths, 663, 646 and 470 nm, against a blank (acetone). The concentrations of chlorophyll a, chlorophyll b, total chlorophyll and total carotenoids were calculated and presented as μg·g$^{-1}$ (dry weight).

The total anthocyanin content was determined according to the pH differential method, and absorbance was measured at 520 and 700 nm. Data were reported as cya-nidin-3-glycoside per 100 g of fw of fruit or 100 g of tissue for at least three replicates [21].

### 2.2.2. GC/FID and GC/MS

The essential oil was analyzed using gas chromatography, performed on an Agilent 7820A GC System, coupled with a flame ionization detector and a 5977B MS detector. The protocol was made according to ISO 3515 [22] for gas chromatographic analysis of lavender oil. The capillary column Econo CapTM ECTM (30 m × 0.32 mm ID, 0.25 μm film thickness) was used. It was operated with an oven program from 40 °C (5 min held) to 240 °C at a rate of 10 °C/min and held at the final temperature for 10 min. Hydrogen (99.999%) was used as a carrier gas at a constant flow rate of 1.0 mL/min. The split ratio was 1:50, the inlet temperature was set to 200 °C and the FID temperature was set to 300 °C.

The GC/MS analysis was performed at all the conditions described above.

The ingredients were quantified by the area of FID peaks without any correction factor. The oil constituents were identified by their mass spectra, matching with the NIST and MS library, and authentic substances were used whenever possible.

### 2.3. Statistics

All chemical data were collected in triplicate and expressed as the mean ± standard deviation. To establish the influence of the type of agricultural system on the essential oil composition and plant pigments on all six farms, statistical procedures were obtained by an ANOVA test for the following factors: type of agriculture system, OF and CF; year, 2019 and 2020 and interaction between type and year. After significant results were obtained by the ANOVA test, Tukey's HSD test was applied to all pairwise differences between means. The significant differences were tested and $p$ values < 0.05 were considered statistically significant. The coefficients of determination $R^2$ were also estimated. The statistical tests were established in XLSTAT 2023.1. 2 (1406).

## 3. Results and Discussion

The essential oil of lavender is deposited in the labial glands and glandular trichomes on the inflorescences. The specifics of the containers require harvesting in dry, sunny and windless weather. The climate data for 2019–2020 showed that temperatures were much higher than the average rates for the winter season (especially 2019). This is the dormancy period for the plants. During the spring months, the temperatures were normal and in June and July they were again higher than normal, but the extent of overshoot was less pronounced. Regarding precipitation, the second period of study, 2020, was characterized by more precipitation than the first period, 2019.

The soil samples in all six arable areas were characterized by acid reactions with pH values ($H_2O$) between 4.2 and 6.4. In the fields with OF the values varied between 4.2 and 5.9—strong acid soil reaction, whereas soil reaction in CF fields was strong to slight acid reaction with pH values between 5.1 and 6.4. With regard to soil organic matter content, higher content in OF fields was found, between 2.11% and 3.78%, than in CF fields where it was lower, between 1.47% and 2.83%.

### 3.1. Content of the Plant Pigments

Chlorophyll contents and pigments determine the color of each leaf, and changes in the amounts of these pigments have been used in various statistical correlations and analyses for inferring plant health, chemistry and physiology [23,24]. Our results about plant pigment concentrations are given in Table 2.

For 2019, Chl a values ranged from 245.9 to 301.0 μg·g$^{-1}$. The average level of CF lavender was 280.7 μg·g$^{-1}$, while for OF it was lower at 272.7 μg·g$^{-1}$. It is also noticeable that the values for OF vary within narrower limits (270.2–275.6 μg·g$^{-1}$), while at CF they are quite variable (245.9–301.0 μg·g$^{-1}$). This indicates the stability of the plants' response to external conditions for OF lavender. Although, for the second year, the values varied within a wider range from 251.3 to 377.5 μg·g$^{-1}$ and the same dependencies were observed. The average chlorophyll a content in OF samples was 267.3 μg·g$^{-1}$—lower than 342.1 μg·g$^{-1}$ for CF. It is not surprising that the application of foliar organic mineral fertilizers in the

budding stage of lavender had a certain positive effect on the chlorophyll content in *L. angustifolia* [23]. At the same time, the levels of variation in OF were significantly closer (251.3–275.7 $\mu g \cdot g^{-1}$) compared with CF (292.8–377.5 $\mu g \cdot g^{-1}$).

**Table 2.** Content of chlorophylls, carotenoids and anthocyanins ($\mu g \cdot g^{-1}$, dry weight) in *L. angustifolia*, grown by OF and CF for the period 2019–2020.

| Farm/Pigments | | Farm 1 (OF) | Farm 2 (OF) | Farm 3 (OF) | Farm 4 (CF) | Farm 5 (CF) | Farm 6 (CF) |
|---|---|---|---|---|---|---|---|
| **Chlorophyll a (Chla)** | 2019 | 275.6 ± 59.1 [f] | 270.2 ± 32.0 [f] | 272.3 ± 17.0 [f] | 295.1 ± 31.5 [f] | 301.0 ± 14.0 [f] | 245.9 ± 24.0 [f] |
| | 2020 | 274.9 ± 39.6 [d] | 275.7 ± 17.8 [d] | 251.3 ± 14.0 [d] | 356.1 ± 74.8 [df] | 377.5 ± 99.0 [df] | 292.8 ± 40.5 [df] |
| **Chlorophyll b (Chlb)** | 2019 | 211.4 ± 54.2 [b] | 193.2 ± 31.3 [b] | 201.7 ± 6.2 [b] | 188.8 ± 50.6 [e] | 240.0 ± 3.8 [e] | 139.4 ± 12 [e] |
| | 2020 | 100.5 ± 52.4 [bde] | 134.6 ± 14.4 [bde] | 137.3 ± 5.2 [bde] | 175.3 ± 25.0 [d] | 217.9 ± 33.9 [d] | 174.8 ± 21.3 [d] |
| **Total chlorophyll** | 2019 | 487.0 ± 113.3 [b] | 463.4 ± 58.7 [b] | 474.0 ± 20.2 [b] | 483.9 ± 76.0 [bd] | 541.0 ± 13.2 [bd] | 385.3 ± 11.4 [bd] |
| | 2020 | 375.3 ± 23.0 [bd] | 410.3 ± 26.3 [bd] | 388.5 ± 18.1 [bd] | 531.4 ± 99.4 [d] | 595.4 ± 131.8 [d] | 467.6 ± 61.8 [d] |
| **Total carotenoids** | 2019 | 52.2 ± 15.5 [abc] | 36.9 ± 9.8 [abc] | 45.9 ± 19.8 [abc] | 61.5 ± 7.8 [c] | 63.1 ± 24.4 [c] | 55.5 ± 7.4 [c] |
| | 2020 | 72.2 ± 32.0 [b] | 70.3 ± 7.7 [b] | 57.5 ± 2.6 [b] | 77.3 ± 15.6 [a] | 65.8 ± 20.0 [a] | 58.5 ± 4.8 [a] |
| **Total anthocyanins** | 2019 | 780 ± 0.1 [ns] | 630 ± 0.5 [ns] | 410 ± 0.2 [ns] | 570 ± 0.2 [ns] | 330 ± 0.3 [ns] | 340 ± 0.0 [ns] |
| | 2020 | 460 ± 0.3 [ns] | 420 ± 0.2 [ns] | 1240 ± 0.6 [ns] | 780 ± 0.1 [ns] | 200 ± 0.0 [ns] | 370 ± 0.0 [ns] |

The data are presented as mean ± SD, [a–f] same superscripts within the same rows (for each parameter) represent significant differences at the level of significance $p < 0.05$; [ns]—non significance.

The values for chlorophyll b during both years fluctuated from 100.5 $\mu g \cdot g^{-1}$ to 240.0 $\mu g \cdot g^{-1}$. The average content for OF lavender was 202.1 $\mu g \cdot g^{-1}$ in 2019, higher than in 2020, in which it was 124.1 $\mu g \cdot g^{-1}$. In contrast, the chlorophyll b concentration for CF was almost the same: 189.4 $\mu g \cdot g^{-1}$ and 189.3 $\mu g \cdot g^{-1}$ for 2019 and 2020.

Maintaining more Chla than Chlb is vital for survival and this ratio is indicative of plant health. Usually, Chla/Chlb varies in the range of 1.5–3.0, where lower values indicate adaptation to stress by activating Chlb, while higher values are maintained under normal conditions [24]. Our data concerning this relation are shown in Figure 4.

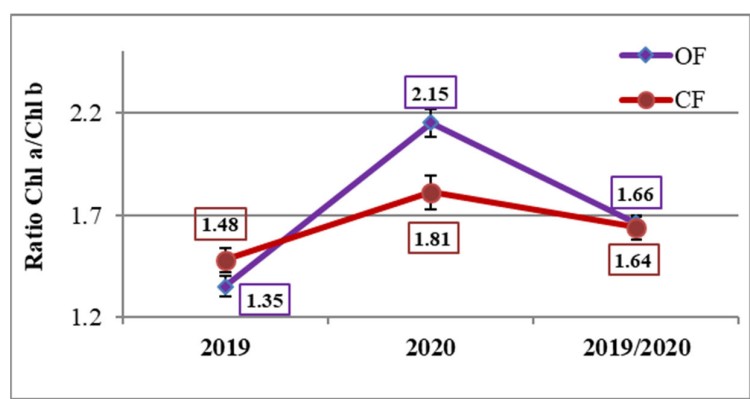

**Figure 4.** Chlophyll a (Chla)/Chlorophyll b(Chlb) content ratio in OF and CF lavender for 2019/2020.

The ratio of Chla/Chlb was between 1.35 and 2.15, which means that the lavender health status on all six farms was good for both OF and CF and both years of study. The lowest value of ratio was for OF during the drier year of 2019, when the Chlb reached the highest concentration of 202.1 $\mu g \cdot g^{-1}$. The average levels concerning both years reveal that lavender has endogenous mechanisms to adapting toward different agricultural systems. More in-depth conclusions can probably be drawn over a longer period of time.

The total chlorophyll content of OF lavender ranged from 375.3 $\mu g \cdot g^{-1}$ to 487.0 $\mu g \cdot g^{-1}$, while for CF these data were in a wider range from 385.3 $\mu g \cdot g^{-1}$ to 595.4 $\mu g \cdot g^{-1}$. The average values for 2019 were higher for OF—474.8 $\mu g \cdot g^{-1}$ than CF—470.1 $\mu g \cdot g^{-1}$. For

the second year, the results obtained were inverse. OF was lower at 391.4 μg·g$^{-1}$ total chlorophyll content and CF was higher with 531.1 μg·g$^{-1}$ TChl. Similar results as ours for the second year were obtained by Ponder and Hallmann [25], who investigated biologically active compounds of raspberry leaves from conventional and organic farming. The authors found higher total chlorophyll content and individual forms of chlorophylls a and b in raspberry leaves from conventional farming than from organic farming. The direct application of mineral elements affects the metabolism and productivity of plant pigments in lavender plants [23]. Chlorophyll levels are comparable to those of nitrogen-nourished *L. angustifolia*, where the content was limited to 1.8–1.9 mg.g$^{-1}$ (dry matter) for control plants and 4.36–5.48 mg.g$^{-1}$ (dry matter) for fertilized ones [26]. A positive correlation between leaf N or N fertilization rate and chlorophyll (Chl) content is well documented for a large number of plant species [27], which explains the higher total chlorophyll content in CF lavender.

Apart from chlorophylls, carotenoids are also important plant pigments, playing a key role in the biosynthesis process. The content of total carotenoids for OF lavender was a minimum of 36.9 μg·g$^{-1}$ and a maximum of 72.2 μg·g$^{-1}$ in the study period. For CF samples, these values were 55.5 μg·g$^{-1}$ and 77.3 μg·g$^{-1}$. The average levels were 55.8 μg·g$^{-1}$ and 63.6 μg·g$^{-1}$, respectively, i.e., 14% higher in CF. The amounts of carotenoids were almost ten times lower than those of chlorophylls, but this is reasonable if one takes into account some evidence that carotenoid accumulation acts as a negative regulation for normal root or shoot development [28], or causes the impairment of normal leaf development and the repression of multiple nuclear and chloroplastic genes [25]. Our data are comparable with the data provided by Biesiada and Kucharska [26]. The influence of the type of agricultural system on the concentration of total carotenoids in organic lavender was established. The average pigment levels of *L. angustifolia* Mill for the two-year study period are presented in Figure 5.

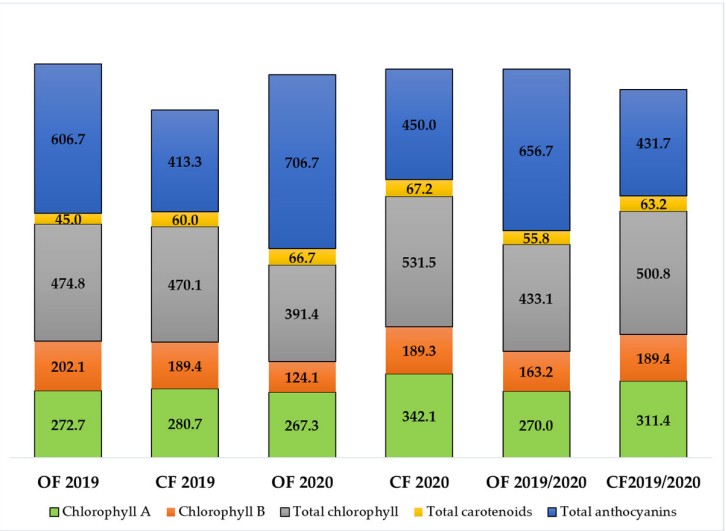

**Figure 5.** Profile of average levels (μg·g$^{-1}$) of plant pigments of *L. angustifolia* Mill in OF and CF for the study period in both systems for each year of study.

The average values of some natural pigments in CF were statistically proven higher than OF only for the second year. For TChl—531.5 and 391.4 μg·g$^{-1}$, Chla—342.1 and 267.3 μg·g$^{-1}$ and Chlb—189.3 and 124.1 μg·g$^{-1}$, with values of $p < 0.01$ and a coefficient of determination $R^2 > 0.58$. This tendency could be explained by the mineral fertilizers for soil and foliar application in conventional farming and its effect in the year with more precipitation, while no additional fertilization was made for organic production. Thus, we could conclude that the influence of the type of agricultural system (OC and CF) on the concentration of chlorophyll a, chlorophyll b and total chlorophyll in lavender *L. angustifolia*

Mill was found for the second year of study, but there is no definite evidence of the influence of this factor in the first year.

It is well-known that most of the polyphenolic constituents (including anthocyanins) vary greatly among individual plant parts, flowers, inflorescence stalks and leaves. The presented results demonstrated that the total monomeric anthocyanin content in *L. angustifolia*, grown by OF and CF, was in the range from 200 to 1240 µg cyn-3-gly.g$^{-1}$ dry weight (Table 2). In general, total monomeric anthocyanins were in the highest content in organically grown lavender samples. Our results were comparable with the reports of Nurzyńska-Wierdak and Zawiślak, who found 0.09% anthocyanins in Polish lavender flowers [29]. It was demonstrated that anthocyanins increased in lavender plants in the examined organs and in leaves they equaled 3.1 mg 100 g$^{-1}$, while in flowers they ranged from 4.3 to 9.9 mg 100 g$^{-1}$ (respectively, flowers in the phase of buds and full development) [29]. In our case, lavender from Bulgaria showed comparable values with Polish lavender samples. The lavender harvest from Farm 5 and Farm 6 (CF) showed the lowest values of total monomeric anthocyanin content, which was comparable with the presence of anthocyanins (0.03%) found in the ethanolic flower extracts of *L. angustifolia* [30] and Romanian lavender flowers 0.047 g cyanidin chloride/100 g herbal product [31]. Moreover, the level of total anthocyanins found by us was comparable with another lavender representative such as *L. spica* L. the flowers of which contained total anthocyanins of 0.4 mg/g [28].

### *3.2. Yield and Chemical Composition of the Essential Oil (EO)*

Data on the chemical composition of OF and CF lavender oils are shown in Table 3.

Under the weather conditions in the study period, oil yields were within normal limits, compared with data for Bulgarian varieties of lavender (1.6–2.6%). This fact confirms that lavender is adaptable to changes in the climate pattern [32]. During the first year, yields ranged from 1.60% to 2.39%. The OF lavender was limited to 1.60–2.29%, while for CF the interval was 1.80–2.39%. The average values for both systems were 2.10% and 2.08%, respectively.

In the second year, oil yields with minimum 1.75% and maximum 2.78% were measured. For OF lavender, the values were between 1.43 and 2.75% and for CF between 1.75 and 2.78%. The average rates were 2.04% and 2.32%, respectively, so the conventional agricultural system had 14% greater yield. The results were higher than the ones reported in [4], where the essential oil content ranged from 1.78% to 2.04% for both systems. High content (2.8% cv. "Lady"—5.0% cv. "Grey Lady") was given for organically cultivated varieties [11]. Other studies compared different lavender cultivars at different geographic locations and practices and EO content oscillated from 0.35% to 2.0% [33] and from 0.2 to 8.1% [34]. There are data [35] about very low content (0.71–1.30%) in dry flowers. Overall, these data confirm the results in [4,12], in which it states that CF lavender has higher yield than OF lavendar, but the statistical procedure by an ANOVA test was not confirmed concerning the influence of the type of cultivation on lavender oil in our study.

The chromatographic profile is typical for the essential oil of *L. angustifolia* [1,5,36], and, more precisely, for the Bulgarian essential oil [22,32,37]. Sixteen compounds have been identified and monitored: the main one was linalyl acetate (28.40–38.23%), followed by linalool (20.01–31.04%), β-caryophyllene (7.92–14.97%), cis-β-ocimene (1.91–12.46%), lavandulyl acetate (2.89–4.55%), terpinen-4-ol (1.62–4.46%) and trans-β ocimene (2.74–3.72%). Different compositions were obtained in different experiments [14,33], which indicates the technological problems or concerns of different lavender subspecies [17].

**Table 3.** Yield of lavender oil, chemical composition of the OF and CF lavender essential oils for 2019/2020 period.

| No | Component, % | Farm 1 (OF) | | Farm 2 (OF) | | Farm 3 (OF) | | Farm 4 (CF) | | Farm 5 (CF) | | Farm 6 (CF) | |
|---|---|---|---|---|---|---|---|---|---|---|---|---|---|
| | | 2019 | 2020 | 2019 | 2020 | 2019 | 2020 | 2019 | 2020 | 2019 | 2020 | 2019 | 2020 |
| | EO, % | 2.22 ± 0.4 ns | 1.94 ± 0.16 ns | 2.29 ± 0.28 ns | 1.43 ± 0.11 ns | 1.80 ± 0.41 ns | 2.75 ± 0.20 ns | 1.60 ± 0.46 ns | 1.75 ± 0.2 ns | 2.26 ± 0.30 ns | 1.75 ± 0.67 ns | 2.39 ± 0.31 ns | 2.78 ± 0.93 ns |
| 1 | α-pinene | 0.25 ± 0.02 ns | 0.62 ± 0.03 ns | 0.27 ± 0.04 ns | 0.24 ± 0.02 ns | 0.31 ± 0.07 ns | 0.19 ± 0.02 ns | 0.32 ± 0.02 ns | 0.33 ± 0.02 ns | 0.16 ± 0.0 ns | 0.23 ± 0.01 ns | 0.13 ± 0.02 ns | 0.14 ± 0.04 ns |
| 2 | Camphen | 0.14 ± 0.02 ns | 0.19 ± 0.02 ns | 0.14 ± 0.02 ns | 0.13 ± 0.00 ns | 0.17 ± 0.02 ns | 0.10 ± 0.00 ns | 0.17 ± 0.02 ns | 0.16 ± 0.00 ns | 0.16 ± 0.02 ns | 0.13 ± 0.02 ns | 0.15 ± 0.00 ns | 0.11 ± 0.02 ns |
| 3 | 3-octanone | 1.04 ± 0.05 ns | 0.98 ± 0.02 ns | 1.11 ± 0.03 ns | 0.86 ± 0.05 ns | 0.95 ± 0.00 ns | 1.48 ± 0.05 ns | 0.29 ± 0.05 ns | 0.29 ± 0.02 ns | 1.87 ± 0.05 ns | 1.18 ± 0.04 ns | 2.44 ± 0.02 ns | 2.91 ± 0.02 ns |
| 4 | Cis-β ocimene | 8.01 ± 0.09 ns | 8.81 ± 0.04 ns | 8.41 ± 0.02 ns | 9.66 ± 0.00 ns | 7.17 ± 0.10 ns | 4.30 ± 0.0 ns | 11.14 ± 0.09 ns | 12.46 ± 0.10 ns | 2.34 ± 0.02 ns | 6.66 ± 0.05 ns | 1.91 ± 0.04 ns | 2.48 ± 0.00 ns |
| 5 | Limonene + 1,8-cineole | 1.45 ± 0.02 ns | 0.91 ± 0.05 ns | 1.57 ± 0.04 ns | 0.92 ± 0.04 ns | 1.36 ± 0.08 ns | 1.30 ± 0.04 ns | 0.91 ± 0.02 ns | 0.47 ± 0.04 ns | 2.54 ± 0.04 ns | 1.09 ± 0.02 ns | 2.69 ± 0.07 ns | 1.84 ± 0.00 ns |
| 6 | Trans-β ocimene | 3.29 ± 0.03 ns | 3.67 ± 0.03 ns | 2.98 ± 0.07 ns | 3.64 ± 0.07 ns | 3.04 ± 0.10 ns | 2.85 ± 0.10 ns | 2.93 ± 0.02 ns | 3.72 ± 0.02 ns | 2.74 ± 0.06 ns | 2.92 ± 0.00 ns | 2.85 ± 0.04 ns | 3.20 ± 0.72 ns |
| 7 | Linalool | 23.65 ± 0.12 ns | 26.83 ± 0.09 ns | 26.62 ± 0.02 ns | 25.50 ± 0.12 ns | 23.16 ± 0.14 ns | 26.49 ± 0.2 ns | 20.01 ± 0.2 ns | 21.55 ± 0.00 ns | 29.64 ± 0.12 ns | 28.03 ± 0.05 ns | 31.04 ± 0.24 ns | 30.54 ± 0.15 ns |
| 8 | Camphor | 0.08 ± 0.00 ns | 0.36 ± 0.06 ns | 0.08 ± 0.02 ns | 0.18 ± 0.04 ns | 0.08 ± 0.01 ns | 0.15 ± 0.05 ns | 0.08 ± 0.01 ns | 0.18 ± 0.01 ns | 0.10 ± 0.02 ns | 0.11 ± 0.02 ns | 0.10 ± 0.02 ns | 0.09 ± 0.02 ns |
| 9 | Borneol | 0.55 ± 0.02 ns | 0.75 ± 0.02 ns | 0.63 ± 0.00 ns | 0.64 ± 0.00 ns | 0.68 ± 0.0 ns | 0.76 ± 0.00 ns | 0.45 ± 0.00 ns | 0.55 ± 0.04 ns | 1.00 ± 0.00 ns | 0.79 ± 0.05 ns | 0.26 ± 0.00 ns | 0.87 ± 0.0 ns |
| 10 | Lavandulol | 0.49 ± 0.15 ns | 0.71 ± 0.00 ns | 0.49 ± 0.00 ns | 0.45 ± 0.00 ns | 0.57 ± 0.00 ns | 0.47 ± 0.02 ns | 0.46 ± 0.02 ns | 0.44 ± 0.02 ns | 0.56 ± 0.00 ns | 0.45 ± 0.05 ns | 0.26 ± 0.02 ns | 0.47 ± 0.0 ns |
| 11 | Terpinen-4-ol | 3.05 ± 0.07 ab | 3.78 ± 0.00 b | 3.83 ± 0.02 ab | 3.56 ± 0.04 b | 3.74 ± 0.02 ab | 3.15 ± 0.03 b | 3.67 ± 0.06 c | 4.46 ± 0.05 ac | 1.62 ± 0.04 c | 3.29 ± 0.09 ac | 1.62 ± 0.07 c | 1.98 ± 0.04 a |
| 12 | α-terpineol | 0.38 ± 0.00 ns | 0.46 ± 0.02 ns | 0.38 ± 0.04 ns | 0.58 ± 0.00 ns | 0.37 ± 0.02 ns | 0.56 ± 0.08 ns | 0.37 ± 0.07 ns | 0.48 ± 0.00 ns | 0.44 ± 0.04 ns | 0.59 ± 0.05 ns | 0.47 ± 0.02 ns | 0.56 ± 0.04 ns |
| 13 | Linalyl acetate | 34.90 ± 0.05 ns | 28.40 ± 0.00 ns | 31.77 ± 0.07 ns | 33.27 ± 0.12 ns | 31.64 ± 0.02 ns | 33.37 ± 0.09 ns | 38.23 ± 0.12 ns | 35.34 ± 0.07 ns | 28.69 ± 0.10 ns | 31.99 ± 0.16 ns | 28.42 ± 0.10 ns | 29.96 ± 0.12 ns |
| 14 | Lavandulyl acetate | 3.89 ± 0.02 ns | 2.89 ± 0.05 ns | 3.41 ± 0.09 ns | 3.68 ± 0.07 ns | 4.55 ± 0.01 ns | 3.71 ± 0.00 ns | 4.16 ± 0.04 ns | 3.70 ± 0.02 ns | 3.93 ± 0.04 ns | 3.52 ± 0.07 ns | 3.88 ± 0.05 ns | 3.44 ± 0.07 ns |
| 15 | β-caryophyllene | 11.43 ± 0.03 b | 8.12 ± 0.07 bd | 10.99 ± 0.10 b | 7.92 ± 0.14 bd | 11.71 ± 0.10 b | 10.73 ± 0.12 bd | 10.05 ± 0.14 cd | 7.95 ± 0.09 cd | 14.97 ± 0.10 cd | 9.40 ± 0.04 cd | 13.65 ± 0.70 cd | 11.19 ± 0.94 c |

The data are presented as mean ± SD, [a–d] same superscripts within the same row represent significant differences at the level of significance $p < 0.05$; [ns]—non significance.

The qualified lavender oil is determined by higher ester content, the ratio of the linalyl acetate/linalool and some minor components, like camphor and terpinen-4-ol [1,37]. Our study revealed that the major ester linalyl acetate fits into the standard, with the exception of two minimum deviations, one for OF and CF samples. According to Konakchiev's [32] study on the strong mobility of esters and flowering stage, these exceptions can be ignored. Another investigation of organic lavender confirms our data with a content of 34.19% [36] and 25–45% [38]. Other sources show extremely low values up to 1.7% [34], even in the absence of the component [4], which is strange and completely contradictory.

In our investigation, the content of linalool was limited in the standard for OF lavender with values from 23.16% to 26.83%. Similar results of organic cultivated lavender were obtained by [37,38]. For our CF oils, this terpene alcohol varied within a wider range and even exceeded the limits of the standard, but reached a maximum level of 31.04%. According to [4], the quantity of linalool oscillated from 42.64% to 59.49% for organic and from 58.74% to 60.02% for conventional lavender oils. Another study reported linalool from 33.3% to 43.4% or 0.1 ÷ 38.7% [34]. The ocimenes are also important for lavender oil quality [1,37]. In our study, the content of cis-β ocimene ranged from 4.30% to 9.6% for OF lavender and from 1.91% to 11.14% for CF oils. The levels of organic samples practically conform to the standard, but for conventional oils, the interval is wider and the values are outside the limits. The content of trans-β ocimene for both types of lavender was a minimum of 2.74% and a maximum of 3.72%, which meets the ISO 3515 requirements.

Oil quality decreases with increasing camphor ratios [1,37] and that is why the compound is restricted to less than 0.6% [22]. In all our samples, both OF and CF, the quantities of camphor are in accordance with the requirements of the standard. In general, there are no differences in the different growing systems for that component. A similar result was reported for organic lavender, where camphor was 0.3% [31]. In an experiment done by [4], the camphor levels were 0.93–4.10% for organic and 0.21–3.76% for conventional lavender. Other studies showed much higher levels of the order of 6.9–14.3% [33] and 0.21–3.76% [34].

Terpinen-4-ol is also an undesirable component that imparts a tart and moldy note to the oil [32,37]. It is also restricted in the international standard. For OF samples, this component varies within very short limits from 3.05% to 3.83% and fits the average standard value. For CF lavender oils, the terpinen-4-ol content is from 1.65% to 4.46%, but practically also confirms the standard.

## 4. Conclusions

It was found that the levels of chlorophyll a, chlorophyll b, and total chlorophyll were higher in conventional farming than in organic ones in the second year of our investigation. Despite clear differences in total anthocyanin content between OF and CF, statistical results did not confirm the influence of the agricultural system on that plant pigment. It was not found the influence of the type of agricultural system on essential oil yields and studied properties in oil composition The case is complex and data serve as a basis for further research on the requirements of lavender plants. Our research will not stop here, because, within such a short period of time, the question of whether the type of conventional or organic system affects the quality of the essential oil cannot be definitively answered, although the initial data show that there is a difference in some plant pigments. Therefore, our monitoring experiment will continue in the same places, and the focus will be expanded on the antioxidant activity of the essential oil obtained from the lavender flower grown in both types of agriculture.

**Author Contributions:** Conceptualization, A.D. and N.G.; methodology, M.G. and M.T.; software, A.D. and N.P.; validation, A.D., N.P. and M.T.; formal analysis, A.D. and N.P.; investigation, A.D., M.T., M.G. and N.P.; resources, M.T. and N.G.; data curation, A.D. and N.P.; writing—original draft preparation, A.D.; writing—review and editing, A.D., M.T. and M.G.; visualization, A.D.; supervision, A.D. and N.G.; project administration, M.T. and Z.Z.; funding acquisition, N.G. All authors have read and agreed to the published version of the manuscript.

**Funding:** This work was supported by the Bulgarian Ministry of Education and Science under the National Research Programme "Healthy Foods for a Strong Bio-Economy and Quality of Life" approved by DCM No 577/17.08.2018; Bulgarian National Science Found, grant agreement No КП-06-Н56/1.

**Data Availability Statement:** The data presented in this study are available on request from the corresponding author. The data are not publicly available due to the studied products are of private manufacturers.

**Acknowledgments:** We appreciate the full cooperation of all the agricultural producers in these municipalities who unconditionally provided their areas for the research.

**Conflicts of Interest:** The authors declare no conflict of interest.

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
