# Peer review of "Organic vs. Conventional Farming of Lavender: Effect on Yield, Phytochemicals and Essential Oil Composition"

_agronomy, doi:10.3390/agronomy14010032_

Round 1
Reviewer 1 Report
Comments and Suggestions for Authors
Comments to the manuscript agronomy-2676107 "Organic vs Conventional Farming of Lavender: Effect on Yield, Phytochemicals and Essential Oil Composition".
Authors propose the report of a two-years experiment of comparison of lavender growing under conventional or organic farming system. Three different orchards held plots under organic treatment and three other plots under conventional growing. Unluckely the six plot were not in the same site and this is clearly evident because differences among sites are higher with respect to the organic or conventional treatments in the results of the experiment. However, Authors assure that plots were in the same environment under homogeneous conditions of climate, soil, cultivar, and general management conditions.
Effects of treatments on oil yield, content of chlorophills and antocyans, and essential oil composition are very small. The experiment demonstrate a substantial lack of quality difference of the oil yield between the two treatments. This is a potential contribute of the research to the field of study and the manuscript may be suitable for publication after some minor changes.
1) Page 2, line 52: please change "4 987" with "4987".
2) Page 2, line 63: please change "nitro-gen" with "nitrogen".
3) Page 3, line 102: please change "L.angustifolia" with "L. angustifolia".
4) Page 4, line 110: are you sure that the annual precipitation is 540 mm? In the Figure 2 the green line "Norms" indicate a rainfall quantity about double.
5) Page 5, line 151: please change "till-age" with "tillage".
6) Page 8, line 266: please check, the 40 micrograms probably are 240 micrograms.
7) Page 8, lines 269-272: please provide a clear reference to these data. The data reported in the Table 2 are quite different.
8) Page 9, line 317: probably the citation number [34] is the number [35].
9) Page 10, Figure 5: please provide the significance level of differences and the result of a mean separation test.
10) The articles listed as number 37 and 38 are not cited in the text.
Comments on the Quality of English LanguageMinor editing of English language required.
Reviewer 2 Report
Comments and Suggestions for Authors
Dear Authors, I would like to extend my gratitude for the opportunity to contribute as a peer reviewer for your valuable work, and I look forward to providing constructive feedback to enhance the quality of this manuscript.
The paper is about an organic (OF) vs. conventional (CF) comparative research applying lavender as plant material and investigating phytochemical and essential oil traits. The basic idea of the experiment is feasible, but the implementation has objectionable parts. I feel a slight bias in the conclusion, which is only partly supported by the results.
General comments:
1. I have fundamental concerns about the basic layout of the experiment. Although I am aware of the difficulties of OF vs CF comparative experimental layouts, I think, that the number of farms within a cultivation method is insufficiently low to represent either OF or CF. The authors must address the issues arising from the differences in microclimatic conditions, soil parameters, cultivation practices, etc. No attempt was made to prove, that these three-three farms can be treated as more or less similar and can be applied in statistical analyses as repetitions or can be used for calculating averages. In the reported datasets I consider differences between farms within the same cultivation methodology in some cases as high (chl a and b for CF in 2020, or anthocyanins for both systems and years, for example), therefore I doubt, that differences in the measured parameters are exclusively due to organic or conventional practices.
2. I can’t see the main difference between the two systems in this layout. According to the provided information, plant protection measures were not applied in both systems, only the form of nutrition was different. Were there any differences in the quantity of applied fertilizers? No information is available about the past of the farms. How long have organic practices been applied, for example? If it is a fresh land right after conversion, it might not make any difference. No information is provided about the origin or the variety of the plant material. Was that the same for all farms?
3. Although I am not a native English speaker, the readability of the ms is very low due to significant grammatical and wording mistakes. The quality of English use is unsatisfactory. Repetitive and irrelevant parts were also present in the ms.
Detailed comments:
line 13-15: In the first sentence the authors must give introduction, not only a sentence containing OF and CF. What is the relevance of this statement to the study?
line 16: nature pigments?
line 18 and later: this part is totally chaotic; I cannot understand why brackets were used for data reporting. Why aren’t they included into the sentence?
line Lamiaceae, and in italics
line 41: What is sustainable tradition? Reword.
line 96: What is the function of this part? Why do you report the results prior to analysis? Is it your preconception?
line 112: reword this sentence.
line 121: What is 0.35-0.40?
line 124: size 5000m2 – what does it mean? Were the fields next to each other? How can you speak about different cultivation methods?
line 132: How did you collect the samples? Elaborate. Were there any rules for it, for example middle of field? I think that one sample cannot represent a 5000 m2 block. I don’t understand how did you get 40 samples? 6 fields * 3 replicates is 18, isn’t it?
Figures: What is norms? Graphs must be self-explanatory. Why is the temperature visualized with columns and the precipitation with consecutive line? It should be reverse.
line 164: what is true lavender?
line 168: end of sentence needs rewording.
line 195 and later: Celsius degree in superscript.
line 232 and 243: Why do you consider non-statistical differences as important? Revise.
Comments on the Quality of English LanguageThe readability of the ms is very low due to significant grammatical and wording mistakes. The quality of English use is unsatisfactory. Repetitive and irrelevant parts were also present in the ms.
Reviewer 3 Report
Comments and Suggestions for Authors
Dear Authors,
thank you for the opportunity to meet the manuscript entitled: "Organic vs Conventional Farming of Lavender: Effect on Yield, Phytochemicals and Essential Oil Composition". The topic of the manuscript is very interesting, especially in relation to the increased pressure to expand the organic approach to plant cultivation on a global level.
The authors focused on the comparison of organic and conventional farming and its influence on the content of oil, pigments, or essential oil components of lavender. Regarding the structure of the manuscript, it follows the requirements of the journal.
However, after studying the manuscript, I have several fundamental comments and recommendations:
Abstract:
L18-24 The abstract should provide a highlight of the most important results. I therefore recommend presenting only the differences between individual approaches and not the ranges for each trait.
Introduction:
Since the main topic of the manuscript is the comparison of organic and conventional cultivation, I recommend expanding the Introduction with the basic aspects of these two approaches and their brief comparison.
L41-42 Interesting information but irrelevant to your research.
L84-86 More suitable for discussion.
Materials and Methods:
One of the basic prerequisites of objective research is the use of a correct experimental design. However, none was used in this case, which I consider a serious flaw.
The use of basic statistical methods to evaluate the experiment is missing. ANOVA will show significance of the influence of the factor on the observed traits. However, it is necessary to use post-hoc analysis to detect differences within the factor.
L115-118 How were the samples taken? In what way and methods were they analysed? Fill in please.
L119 It would be appropriate to specify nutrient contents.
L124 "...size of experimental block..." Not sure if this is the correct label. Experimental block is a continuous area, but in your case it is divided into several parcels in other locations. I recommend reconsidering this label.
L132-134 The sampling method has already been specified above. It is not clear why there were 40 samples altogether.
L148 This is confusing information. It makes a fundamental difference whether 20 or 30 t/ha of manure is applied. This can cause significant differences in results. Experiments must have a clear methodology.
L163 Where did the plant material come from? The name of the variety used and the basic characteristics are desirable.
There is no information on the harvest date and the way in which the collection was carried out.
Results and discussion
The results are only commented on in a very general way and often do not focus on important trends.
E.g.: Looking at Figure 3, the result of OF Yasenovo in 2020 is interesting. This result also had a significant impact on the averages evaluated in the text. However, the authors missed the opportunity to discuss what might have been the reason for the extreme increase.
L232 "There is no statistically proven difference..." On what basis was this conclusion made?
L244 "...not found significant statistical difference between the organic and conventional agricultural systems." By what method was this evaluated?
Table 2 Row 6 (Krun) When evaluating experiments, it is always beneficial to attend to and discuss extreme results. How do the authors explain the results on this site? They absolutely do not correspond with other CFs.
There are a number of studies that address the effect of organic and conventional fertilization on chlorophyll content. However, the authors did not discuss this topic at all, while the influence of mineral fertilization on pigments is clear.
Conclusion
The conclusions are brief and do not fully follow the results of the experiment. Need to expand.
Round 2
Reviewer 2 Report
Comments and Suggestions for Authors
Dear Authors, thank you for your answers. However, I did not get a response to my fundamental concern, therefore I copy it here again.
General comments:
1. I have fundamental concerns about the basic layout of the experiment. Although I am aware of the difficulties of OF vs CF comparative experimental layouts, I think, that the number of farms within a cultivation method is insufficiently low to represent either OF or CF. The authors must address the issues arising from the differences in microclimatic conditions, soil parameters, cultivation practices, etc. No attempt was made to prove, that these three-three farms can be treated as more or less similar and can be applied in statistical analyses as repetitions or can be used for calculating averages. In the reported datasets I consider differences between farms within the same cultivation methodology in some cases as high (chl a and b for CF in 2020, or anthocyanins for both systems and years, for example), therefore I doubt, that differences in the measured parameters are exclusively due to organic or conventional practices.
2. I can’t see the main difference between the two systems in this layout. According to the provided information, plant protection measures were not applied in both systems, only the form of nutrition was different. Were there any differences in the quantity of applied fertilizers? No information is available about the past of the farms. How long have organic practices been applied, for example? If it is a fresh land right after conversion, it might not make any difference. No information is provided about the origin or the variety of the plant material. Was that the same for all farms?
Comments on the Quality of English LanguageI don't see general improvements on English use.
Reviewer 3 Report
Comments and Suggestions for Authors
The authors made considerable efforts to improve the quality of the manuscript. Thanks for the replies to my comments. However, there are still many questions regarding the methodical handling of the experiment. It is possible to assume that non-homogeneous conditions (different nutrition, varieties, etc.) caused fundamental differences in the results. Anova was used to evaluate the influence of the factor, but I consider it necessary to also evaluate the evidence within the factor. I still consider these to be basic limits for publishing.
